# Deep activity propagation via weight initialization in spiking neural networks

**Aurora Micheli**
Delft University of Technology
a.micheli@tudelft.nl

**Olaf Booij**
Delft University of Technology
o.booij@tudelft.nl

**Jan van Gemert**
Delft University of Technology
j.c.vangemert@tudelft.nl

**Nergis Tömen**
Delft University of Technology
n.tomen@tudelft.nl

## Abstract

Spiking Neural Networks (SNNs) offer advantages such as sparsity and ultra-low power consumption, making them a promising alternative to conventional neural networks (ANNs). However, training deep SNNs is challenging due to the quantization of membrane potentials into binary spikes, which can cause information loss and vanishing spikes in deeper layers. Traditional weight initialization methods from ANNs are often used in SNNs without accounting for their distinct computational properties. In this work, we derive an optimal weight initialization method tailored for SNNs, specifically taking into account the quantization operation. We demonstrate through theoretical analysis and simulations with up to 100 layers that our method enables the propagation of activity in deep SNNs without loss of spikes. Experiments on MNIST confirm that the proposed initialization scheme leads to higher accuracy, faster convergence, and robustness against variations in network and neuron hyperparameters.

## 1 Introduction

Spiking Neural Networks (SNNs) are a class of artificial neural networks inspired by the dynamics of biological brains, where information is encoded and transmitted through discrete spikes [1, 2, 3]. This unique mode of communication enables SNNs to perform fast computations with low power consumption [4, 5], especially when combined with specialized neuromorphic hardware [6, 7, 8]. However, SNNs still underperform compared to conventional artificial neural networks (ANNs), mainly due to the additional challenges associated with their training. While ANNs are typically trained via gradient descent, the discrete nature of spikes in SNNs complicates the use of backpropagation. Different methods such as ANN-SNN conversion [9, 10, 11] and backpropagation with surrogate functions [12, 13] have been proposed to circumvent this problem. Nevertheless, the proposed solutions haven't been sufficient to fully bridge the performance gap between SNNs and ANNs, without compromising the efficiency advantages of SNNs.

Similarly to ANNs, in SNNs a suboptimal weight initialization can hamper the training process [14, 15], especially in deep networks [16]. While ANN weight initialization strategies tailored to specific activation functions and weight distributions have been widely explored [17, 18, 19], these methods are also often inappropriately applied to SNNs. Unlike ANNs, SNNs feature temporal dynamics, resetting mechanisms, information quantization and their activation function differs from those examined in the ANN literature. Hence, ANN initalization schemes are inadequate for SNNs and often cause undesired effects such as vanishing or exploding spikes in deeper layers.

In this paper, we analytically derive a weight initialization method that accounts for the specific activation function of Spiking Neural Networks (SNNs), building on the approach proposed in [18]

38th Second Workshop on Machine Learning with New Compute Paradigms at NeurIPS 2024(MLNCP 2024).

for standard ANNs. We empirically demonstrate that, unlike the standard ReLU-based method, our initialization enables spiking activity to propagate through deep networks without dissipation or amplification. Additionally, we show that proper weight initialization leads to improved accuracy, faster convergence, and lower latency in a simple classification task.

## 2 Related work

A proper initialization method should avoid reducing or amplifying the magnitudes of input signals. In ANNs, Glorot & Bengio in [17] address the issue of saturated units for the logistic sigmoid activation function and propose a new weight initialization scheme aimed at maintaining constant activations and gradient variance across layers. He et. al [18] extend this analysis to Rectified Linear Unit (ReLU) activations, introducing the widely adopted Kaiming initialization for deep ANNs.

In contrast, research on initialization schemes for SNNs is scarce and the problem of information propagation in SNNs has been often indirectly addressed. In [20] and [21] the appropriate membrane leak factor and firing threshold are learnt during training. Similarly, in [22, 23] some learnable parameters are incorporated in the SNN to optimize the neuron firing rate, thus increasing the computational complexity. Additionally, approaches like global unsupervised firing rate normalization [24], batch normalization adapted to SNNs [25, 26] and constraints on the membrane potential distributions [27, 28] help regulate spike response and information flow.

Some studies regard ANN-SNN conversion as an initialization method [29], but this is limited to rate-based encoding, restricting its applicability to networks with alternative encoding schemes.

Only a few studies directly address what constitutes a good initial state for SNNs. Some works attempt to empirically determine a suitable weight scale in the case of SNNs, often lacking a solid theoretical foundation [16, 30, 31, 32]. In [33] the authors derive a new initialization strategy considering the asymptotic spiking response given a mean-driven input. [15] proposes a fluctuation-driven initialization scheme, but neglects both the spiking and the resetting mechanism. In [34] a specular approach similar to the one presented in [18] is studied, yet the theoretical insights lack empirical validation.

## 3 Methods

### 3.1 The spiking neuron

The Leaky-Integrate-and-Fire (LIF) neuron [1] is one of the most popular models used in SNNs [2, 35] and neuromorphic hardware [7, 6] to emulate the functionality of biological neurons. The state of a LIF neuron at time $t$ is given by the membrane potential $U(t)$ which evolves according to

$$\tau \frac{dU(t)}{dt} = -U(t) + RI(t), \tag{1}$$

where $\tau$ is the membrane time constant, $R$ is the resistance of the membrane and $I(t)$ is the time-varying input current. Following previous works [26] we use the discretized equation with time step $\Delta t$ which gives membrane potential $u$ at time step $t$ as:

$$u^t = \beta u^{t-1} + \sum_j w_j x_j^t, \tag{2}$$

where $\beta \propto (1 - \Delta t/\tau)$ is a leak factor $\in [0, 1]$ governing the rate at which the membrane potential decays over time, $j$ is the index of the pre-synaptic neuron, $w_j$ represents the weight of the connection between the pre- and post-synaptic neurons and $x_j$ is the binary spike activation. When the membrane potential $u$ exceeds a firing threshold $\theta$, the neuron emits a binary output spike $x = 1$. After firing, the membrane potential is reset by subtracting from its value the threshold $\theta$ (soft reset).

### 3.2 Weight initialization for a spiking neural network

Our derivation is inspired by He et al. [18], which suggests that an effective weight initialization should enable information flow across many network layers by keeping the variance of the input to each layer constant. We examine the variance of responses within each layer of a fully-connected SNN initialized at time step $t = 0$.

For a generic layer $l$ with $m$ neurons:

$$\boldsymbol{u}_l = \boldsymbol{w}_l \boldsymbol{x}_l \tag{3}$$

$$\boldsymbol{x}_l = f(\boldsymbol{u}_{l-1}) \tag{4}$$

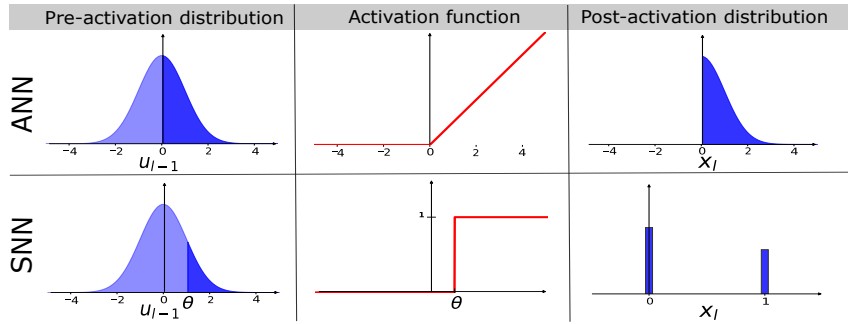

Figure 1: Comparison of standard activation functions for ANNs (*top*) and SNNs (*bottom*). When applied to pre-activation distribution $u_{l-1}$ (*left*) the SNN thresholding mechanism (*middle*) generates binarized activations $x_l$ (*right*). The dark shaded areas of $u_{l-1}$ correspond to the fraction of neurons which will be activated and provide non-zero input to the next layer. With identical input distributions, this fraction is considerably lower for SNNs. This highlights why weight initializations optimized for ReLU will lead to vanishing activity in deep SNNs.

Here $\boldsymbol{x}_l \in \{0, 1\}^n$ is a binary vector representing the $n$ input spikes, $\boldsymbol{w}_l \in \mathbb{R}^{m \times n}$ is the weight matrix and $\boldsymbol{u}_l \in \mathbb{R}^m$ represents the membrane potentials of neurons in layer $l$. $\boldsymbol{x}_l$ is obtained by applying the activation function $f$ to the membrane potentials of layer $l-1$. In a conventional SNN $f$ is defined as the Heaviside step function:

$$f(\boldsymbol{u}_{l-1}) = \begin{cases} 1, & \text{if } u_{l-1} > \theta \\ 0, & \text{if } u_{l-1} < \theta \end{cases} \tag{5}$$

where $u_{l-1}$ are the elements of $\boldsymbol{u}_{l-1}$ and $\theta > 0$ is the neurons firing threshold. We assume that the elements of $\boldsymbol{w}_l$ are mutually independent and share the same distribution (i.i.d.). Following [18] and [17], the elements of $\boldsymbol{x}_l$ are also considered to be mutually independent and identically distributed (i.i.d.). Lastly, $\boldsymbol{w}_l$ and $\boldsymbol{x}_l$ are independent of each other. We can then write:

$$\text{Var}[u_l] = n_l \text{Var}[w_l x_l]. \tag{6}$$

Here $u_l$, $w_l$, and $x_l$ represent each random variable element in $\mathbf{u}_l$, $\mathbf{w}_l$ and $\mathbf{x}_l$ respectively. We choose $w_l$ to be symmetrically distributed around 0. Since $w_l$ and $x_l$ are independent of each other, we can rewrite the variance of their product as:

$$\text{Var}[u_l] = n_l \text{Var}[w_l] E[x_l^2], \tag{7}$$

where $E[x_l^2]$ is the expected value of $x_l^2$. It is worth noting that the expression $E[x_l^2]$ strongly depends on the network activation function. Here is where our derivations differ from He et al. [18].

By assuming that $u_{l-1}$ is zero-centered and symmetric around its mean, for a ReLU activation function, $x_l = \max(0, u_{l-1})$, one obtains $E[x_l^2] = \frac{1}{2}\text{Var}[u_{l-1}]$. This result stems from the fact that the ReLU function preserves exactly the positive half of the distribution it acts upon. As depicted in Figure 1, this doesn't hold true for the activation function of SNNs where, by definition, $\theta > 0$. This difference leads to considerably sparser activations in SNNs. In the case of an SNN, we can express $E[x_l^2]$ as:

$$E[x_l^2] = \sum_{j=1}^n x_l^{j^2} P(x_l = x_l^j). \tag{8}$$

The binary elements $x_l^j \in \{0, 1\}$ represent spikes. Applying the SNN activation function (5) to Eq. 8, we find that $E[x_l^2] = P(u_{l-1} > \theta)$. Equation 7 can then be rewritten as:

$$\text{Var}[u_l] = n_l \text{Var}[w_l] P(u_{l-1} > \theta). \tag{9}$$

As commonly done in recent works [28, 20, 36, 23], we consider a real-valued input $I_0$ encoded to binary spikes using the first layer of the SNN. When feeding $I_0$ to the membrane potentials $u_0$ of the initial layer, $u_0 = I_0$ and $u_0$ trivially follows the same distribution as the input. We let $I_0$ be standard normal distributed $I_0 \sim \mathcal{N}(\mu = 0, \sigma^2 = 1)$, thus $\text{Var}[u_0] = 1$, $E[u_0] = 0$. A proper initialization method should avoid reducing or amplifying the magnitudes of the input signals when propagated

across the network layers. This condition can be met if $\text{Var}[u_l] = 1$ for every layer $l$, which lets us simplify Eq. 9 and leads to a zero-mean Gaussian weight distribution with variance:

$$\text{Var}[w_l] = \frac{1}{n_l P(u_{l-1} > \theta)} \tag{10}$$

Equation 10 is our proposed weight initialization method for training deep SNNs. Note that in terms of architecture parameters, it only depends on the number of input neurons $n$.

Because $u_{l-1}$ is symmetric around 0 and $\theta > 0$, then $P(u_{l-1} > \theta) < \frac{1}{2}$. It is important to note that:

$$\frac{1}{n_l P(u_{l-1} > \theta)} > \frac{2}{n_l}, \tag{11}$$

Where $\frac{2}{n_l}$ is the standard initialization for a ReLU network [18]. Thus, initializing the weights of an SNN using a method designed for conventional ANNs with ReLU activation functions does not ensure the propagation of information from the input throughout the network.

## 4  Validation with numerical simulations

Unless otherwise specified, we consider fully-connected SNNs with 100 layers and $n = 1000$ LIF neurons in each layer. The input $I_0$ is real-valued and randomly drawn from $\mathcal{N}(\mu = 0, \sigma^2 = 1)$. Consistently with the derivation in 3.2, we encode the inputs to binary spikes by feeding them to the membrane potentials of the initial LIF layer $u_0$.

We investigate the behavior of activity propagation under different weight initialization schemes and compare our method against the prevailing choice for conventional ANNs: Kaiming initialization ([18]). The weights in the network are therefore randomly initialized respectively from $\mathcal{N}(0, \sqrt{\frac{1}{nP(u_0 > \theta)}})$ (our method) and $\mathcal{N}(0, \sqrt{\frac{2}{n}})$ (Kaiming), where $n$ is the layer width. Since $u_0 \sim \mathcal{N}(0, 1)$, $P(u_0 > \theta)$ is defined as:

$$P(u_0 > \theta) = \int_\theta^\infty \frac{1}{\sqrt{2\pi}} e^{-\frac{u_0^2}{2}} du_0. \tag{12}$$

The integral in Eq. 12 doesn't have a closed-form solution, but it can be numerically estimated using the error function [37]. We recall from 3.2 that, to retain the activity over depth, we aim to conserve $\text{Var}[u_l]$ across the layers. We initialize the network at time $t = 0$, propagate the input across its layers, record the values of the membrane potentials $u_l$ in every layer $l$ and compute their variance. Figure 2 shows how $\text{Var}[u_l]$ evolves with depth for the 2 different initialization schemes and for 6 different values of the firing threshold $\theta$. For every different value of $\theta$, we run the simulation 20 times and plot the average. The shaded areas represent the standard deviation over the different runs.

The results (Figure 2, left) demonstrate that in an SNN initialized with our proposed method, the variance $\text{Var}[u_l]$ of the neuron states stays constant across layers regardless of the threshold $\theta$, as the theory predicts. Conversely, with Kaiming initialization, information dissipates across layers, especially as firing thresholds increase. The only effective way to preserve information with Kaiming is to set $\theta = 0$, where the activation function becomes effectively equivalent to ReLU.

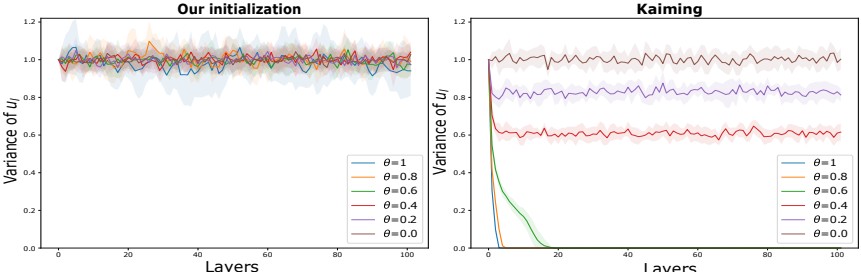

Figure 2: Propagation of $\text{Var}[u_l]$ across network layers for (*left*) our initialization scheme and (*right*) Kaiming for six firing threshold values ($\theta$). For all $\theta$, our proposed initialization method enables information propagation across all 100 layers. In contrast, Kaiming initialization leads to information dissipation across layers, particularly evident with higher threshold values. Each simulation was repeated 20 times, and the shaded areas represent the standard deviation over these runs.

## 4.1 Extension to multiple time steps

SNNs excel at processing time-dependent input thanks to the intrinsic memory of their spiking neurons, represented by the membrane potential $u$. This makes them particularly suited for tasks involving dynamic temporal patterns, like speech recognition and video analysis [38, 39, 40].

In this section, we extend the analysis from Section 4 to examine how weight initialization impacts information propagation across both space *and* time in multiple time-step simulations of deep SNNs. We employ fully-connected SNNs of 100 layers with $n = 1000$ neurons in each layer. We consider LIF neurons with soft reset and numerically compute the discrete-time dynamics based on Eq. 2. The dynamics of the membrane potentials including the reset term is given by:

$$u_l^t = w_l^t x_l^t + \beta u_l^{t-1} - x_{l+1}^{t-1}\theta \tag{13}$$

for time step $t > 0$ and layer $l$. $\beta \in [0, 1]$ is the leak factor. Network weights are randomly initialized either using our initialization scheme or Kaiming, and the inputs are randomly drawn from $\sim \mathcal{N}(0, 1)$, same as in Section 4. Differently, in this section, we iteratively feed the input (constant over time) to the membrane potentials of the initial LIF layer $u_0^t$ at every time step $t$. We compute the variance of the membrane potentials $u_l^t$ and the total number of spikes at every layer $l$ and time step $t$, for a total of $T = 20$ time steps. We repeat each simulation 10 times, and report their average.

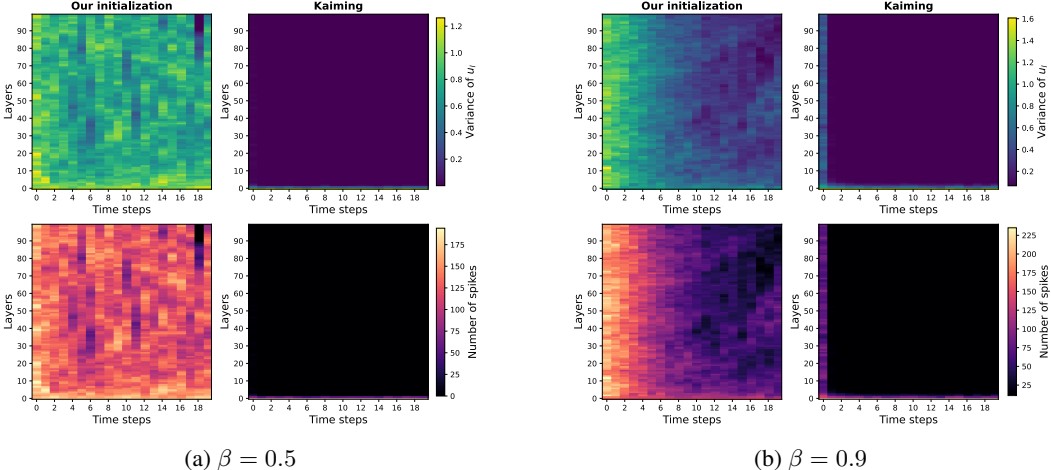

(a) $\beta = 0.5$          (b) $\beta = 0.9$

Figure 3: Propagation of $\mathrm{Var}[u_l^t]$ (*top row*) and number of spikes (*bottom row*) across layers and time steps for our initialization method and Kaiming averaged over 10 runs. **(a)** Our proposed weight initialization preserves activity and propagates spikes through 100 layers and 20 time steps. In contrast, with Kaiming initialization neuronal activity rapidly dies out. **(b)** The effect of the leak and reset terms becomes more pronounced for high values of $\beta$ and pushes the network into the dissipative regime. However, with our proposed initialization method, we can still successfully retrieve an output.

The network initialized with our method succeeds in conserving the $\mathrm{Var}[u_l^t]$ across both space and time, whereas the network initialized with Kaiming fails (Fig. 3a). Conserving $\mathrm{Var}[u_l^t]$ is crucial, as it means conserving the number of spikes, and therefore ensuring a consistent network output.

Although our mathematical derivation does not explicitly take time into account, unlike methods derived for ANNs, it considers the specific SNN activation function, and by keeping the variance of the membrane potentials $u_l$ constant, it aims to indirectly keep the variance of the layer input $w_l x_l$ constant (Eq. 3). This helps to effectively propagate information also across multiple time steps. Nevertheless, deviations from theory are expected due to the leak and reset terms (Eq. 13). In particular, the reset operation affects the membrane potential distributions, violating the assumption of a normal distribution symmetrically centered around 0 (see Appendix 7: Figure 5). However, how well the normal distribution still holds as an approximation depends on neuron hyperparameters. For example, deviations from theory are visible at higher values of $\beta$. A larger $\beta$ leads to broader distributions of $u_l^t$, and thus to a more abrupt change in the distributions when neurons with $u_l^t > \theta$ are reset. As illustrated in Fig. 3b, when $\beta = 0.9$, the network dissipates energy over time. We attribute this dissipation to the shift in $u_l^t$ distributions. Still, we note that with our proposed initialization method, we can still successfully retrieve an output, unlike with Kaiming.

# 5 Experiments on MNIST

To empirically evaluate our variance-conserving weight initialization, we conduct object classification experiments using the MNIST digits dataset [41]. The 28×28 pixel grey-scale images are normalized to have mean 0 and variance 1, in line with the assumptions used in the derivations. As in Section 4 the inputs are encoded to binary spikes using the first LIF layer. The final layer of the network outputs binary spikes, which are accumulated over time steps and passed to the cross-entropy loss function. Unless otherwise specified, we employ a fully-connected SNN consisting of 10 layers, each comprising $n = 600$ LIF neurons with soft reset. We set $\theta = 1$ and $\beta = 0.5$.

Commonly, SNNs performing spike-count based object classification use a large number of total time steps $T$. A typical range of $T$ can be between 10 and a few thousand [42]. Here, we set the number of total time steps to $T = 3$. We hypothesize that initializations which enable constant information propagation across depth might also enable inference with low latency, where there is no need to wait for many time steps to accumulate the necessary number of output spikes.

The network is trained for 150 epochs using backpropagation through time (BPTT) [16] and the arctan surrogate gradient function [23]. We utilize the Adam optimizer [43] with a learning rate of $1 \times 10^{-3}$ and employ cosine annealing scheduling. The runtime for each experiment is approximately 2 hours on a single GPU. Our weight initialization method is compared to Kaiming, as in previous sections.

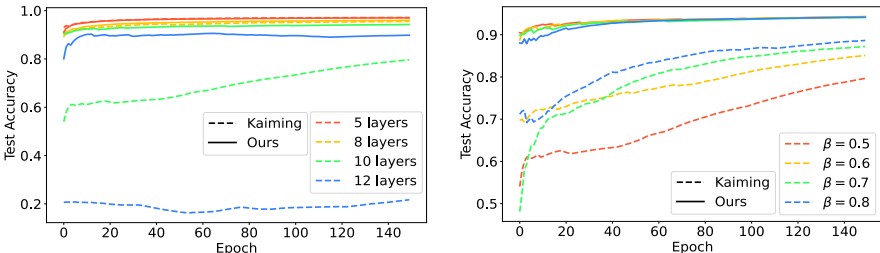

Figure 4: Test accuracy on MNIST for different values of network depth (*left*) and $\beta$ (*right*). Our proposed initialization (solid lines) achieves better generalization than Kaiming (dashed lines).

The results illustrated in Figure 4 show that our method generally allows an SNN to converge faster and to achieve better accuracy than Kaiming for different values of network depth and $\beta$. We attribute the poor convergence and lower task performance of Kaiming-initialized networks to inadequate activity propagation, as supported by both theoretical insights and empirical findings from previous sections. Specifically, training becomes increasingly challenging with deeper networks, lower values of $\beta$ (leading to less information retention from the previous time step) and higher values of $\theta$ (resulting in fewer neurons emitting spikes) (see Appendix 7: Figure 6).

# 6 Conclusion and Discussion

In this paper, we address the problem of weight initialization in Spiking Neural Networks (SNNs) and show how the techniques developed for ANNs, such as Kaiming initialization, are inadequate for SNNs. We analytically derive and empirically test a novel weight initialization method which takes into account the specific activation function of SNNs. Our weight initialization depends only on the number of input neurons $n$ to a layer and is therefore broadly applicable to all deep, spiking network architectures with fixed connectivity maps. We demonstrate that our proposed initialization is robust against variations in several network and neuron hyperparameters, which can enable deep activity propagation for diverse models and machine learning tasks.

A limitation of our proposed initialization is that it does not account for the temporal dynamics of membrane potentials $u_l$. Specifically, after neurons are reset, our assumption that $u_l$ is normally distributed around zero is violated. We observe that the extent to which the normal distribution remains a valid approximation depends on the neuron hyperparameters, although our initialization seems more robust than Kaiming and the theory can be expanded to explicitly take into account the temporal variations in $u_l$. Another assumption of our derivation is that the activations $x_l$ are mutually independent. This assumption is typically violated in the case of real-world data, such as images. Empirically, in section 5 we illustrate how, for an SNN trained on MNIST, our variance-conserving initialization scheme still translates into accelerated training, improved accuracy and low latency compared to Kaiming. Nevertheless, we acknowledge the necessity of extending this analysis to more complex architectures and datasets, in order to evaluate its effectiveness in various settings.

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

# 7   Appendix

The reset operation violates the assumption that $u_l^t$ is always normally distributed and symmetrically centered around 0, especially for higher values of $\beta$. In Figure 5 we show the values of skewness and excess kurtosis for $u_l^t$ across layers and time steps in the case of $\beta = 0.9$. Skewness measures the degree of asymmetry of the distribution, while excess kurtosis measures the degree of peakedness and flatness of a distribution. A normal distribution has 0 skewness and 0 excess kurtosis. We note how $u_l^t$ tends to a left-skewed and heavy-tailed distribution.

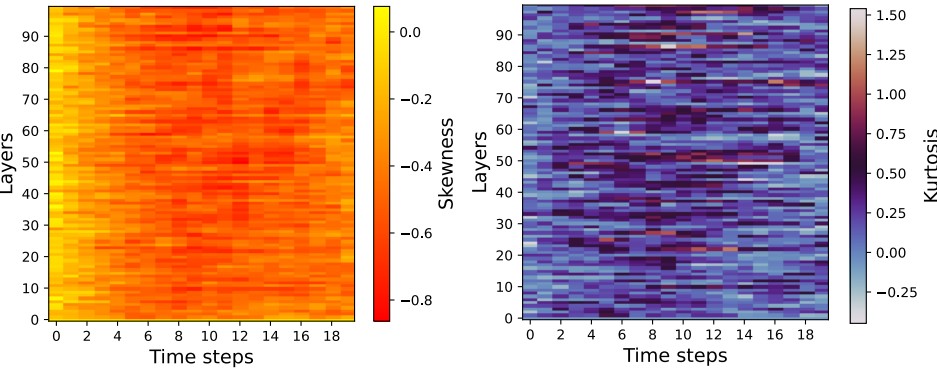

Figure 5: Skewness (*left*) and excess kurtosis (*right*) of $u_l^t$ across layers and time steps for $\beta = 0.9$.

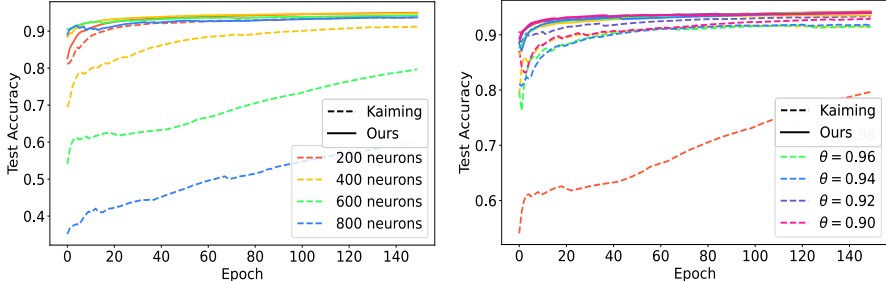

Figure 6: Test accuracy on MNIST for different values of layer width (*left*) and $\theta$ (*right*). We compare our proposed initialization method (solid lines) to Kaiming (dashed lines) and find that it achieves better training accuracy and faster convergence.

