# OpenReview forum: "Deep activity propagation via weight initialization in spiking neural networks"
_NeurIPS.cc/2024/Workshop/MLNCP — MLNCP Poster_

### Official Review · Reviewer_kgQ6 · 2024-09-30
**Learning SNN**

**Rating:** 7
**Confidence:** 4

**Review:**

SNNs, as a promising alternative to ANNs, have the advantages of low power consumption and sparsity. However, in deep SNNs, due to the nature of binary spikes, the propagation process leads to information loss thus making their training and application challenging. The initialization of network weights is very important for the propagation of information in deep networks, and currently in the field of SNNs, it is common to use the initialization methods of ANNs directly. However, the initialization methods of ANNs often consider the role of continuous activation functions, and the discrete, special activation function with temporal properties used in SNNs make the ANNs initialization methods not sufficient to solve the problem of information vanishing and explosion in SNNs. To solve the above problem, this work proposes a new weight initialization method for SNNs. The proposed method takes into account the nature of using discrete spikes in SNNs, by using the probability of neurons being fired, to regulate the initialized weights. The experimental results show that in the cases of deep SNNs with multiple time steps, the proposed method allows the information to be propagated efficiently without too much loss, whereas the initialization method of ANNs leads to severe information vanishing problem when applied on SNNs.
I think the work is meaningful and the derivation process and experimental results are convincing. However I think the authors should have gone into more detail in the results section. Three are the main points I refer:
1. In the section “Validation with numerical simulations”, please describe in detail the settings of the SNNs used, such as the time step, and calculate the value of the P(u > \theta).
2. In the section “Experiments on MNIST”, the authors mention that T in the conventional case belongs to tens to thousands, but the study uses T=3 and assumes that the proposed method in the conventional case (10<T<1000) still works as well as there is no need to wait for long time steps. Please explain the reason for choosing T=3. How about the performance of conventional cases?
3. In the section “Experiments on MNIST”, the authors mention that the proposed method yields better accuracy and faster convergence rate. Please add tables to summarize the performances, so we can compare them clearly. From the current figure 4, when the number of layers is less than 8, the comparison of convergence speed and accuracy is not clear, please add specific values.
Another point is the theoretical derivation as well as the experimental proofs use the case where the input information is normalized to have a variance of 1 and a mean of 0. Does the derivation process of the method need to be adjusted when the distribution of the inputs is set to something else?
Minor points:
In fig.4 (right), what is the setting of the used model? The labels should be different with fig.4(left), it is a little confusing.
In no.188 line, “training becomes increasingly challenging with deeper networks ”, in the appendix, only results of number of nodes are shown. There is no description of number of layers.
In the appendix, fig.6 (right) has 6 lines, but only 4 labels, and what are their settings?

---

### Official Review · Reviewer_wXas · 2024-10-04
**The authors derive a weight initialization method for an SNN based on its activity in one single time step and instantaneous spike propagation. They show on an MNIST task, that their method outperforms Kaiming initialization.**

**Rating:** 4
**Confidence:** 5

**Review:**

The manuscript provides a new weight initialization method for SNNs, but lacks both, a solid motivation on what are the exact gaps they address with their theory to improve over existing SNN initialization methods (which already address methods similar to Kaiminig initialization for the spiking non-linearity) and a simulation-based comparison showing how they improved over existing SNN initialization methods.
Hence, I don't see what the manuscript adds over existing initialization methods for SNNs.

Moreover, there are other points, that should be addressed to improve clarity and relevance of the manuscript.

1. Chapter 3.2: while it looks like a good start to consider initially only the first time step, this in fact reduces the SNN to a binary perceptron, not considering any of the neuron dynamics or the reset. Could you comment on why you chose on that approach and what are the advantage of this approach compared to other taken approaches? Moreover, it would be nice if you make an effort towards extending your theory to multiple time steps or at least discuss the theoretical implications in chapter 4.1 as well.

2. Fig. 1 provides a nice intuition, unfortunately, the captions and labels are not readable.

3. While you comment quickly at the end of chapter 3 on how your initialization compares to Kaiming, it would be nice to see how it compares to other initialization methods that were developed for SNNs, since they were specifically developed for the same spiking activation function.

4. Again, also for the empirical validation in chapter 4, it would be nice to see comparison to other SNN initialization methods, as you wanted to improve over them.

4. Furthermore, I was wondering, why the threshold at 1 is so important, as this is a rather arbitrarily chosen value for an SNN and could as well be at another value.

5. In chapter 4.1, you wanted to extend your initialization method to multiple time steps, however, there is no theory on it and no discussions on the approximations that would be necessary for such an extension.

6. I would consider 20 time steps still rather short, if you want to check, how time, and hence dynamics and reset influence the network behaviour. While the spike propagation is somewhat convincing for Fig. 3a, they do so at the cost of biological plausibility or reduced dynamics, i.e. the exponential decay of the membrane potential cannot be captured properly, if the membrane time constant is only twice the timestep, as $\beta=0.5$ means. Moreover, for $\beta=0.9$, which I assume the more realistic setting, one can see a quite prominent decay in the number of spikes over time steps. How does this evolve over more time steps, where the full dynamics are captured?

7. From the left plot on Figure 4, it looks like Kaiming init with 5 layers performs best, i.e. better than your init. What is your take on that?
8. Again Fig. 4, did you average these results over multiple initializations with different seeds? It would be nice to see the mean and the standard deviation, which would make the results more reliable.
9. Again Fig. 4. it seems, that for the left plot, you chose $\beta=0.5$, which is good for your theory but worst for Kaiming (see right) and less biologically realistic than $\beta=0.9$. Again, for the right plot, you chose 10 hidden layers, where your init performs well, but Kaiming doesn't (while for 5 layers, it is best). Why did you make these choices? And why do you need so many layers to solve MNIST?
10. Also related MNIST training, is the variance still conserved over layers?
11. Again MNIST training, did you use regularizers?
12. Again MNIST, you say in line 175 that typical number of time steps are between 10 and a few thousands, where I feel, that around 100 it starts to become more biologically realistic. Why did you use only 3 timesteps and not more?

13. Importantly, you are missing a comparison to existing SNN initialization methods.

14. The colors in Fig. 6 don't match the legend.

---

### Decision · Program_Chairs · 2024-10-10

Accept (Poster)